# Mothers of children with major congenital anomalies have increased health care utilization over a 20-year post-birth time horizon

**Nirav R. Shah**[1�]*, **Kyung Mi Kim**[1�], **Venus Wong**[1], **Eyal Cohen**[2,3‡], **Sarah Rosenbaum**[1], **Eli M. Cahan**[1,4], **Arnold Milstein**[1‡], **Henrik Toft Sørensen**[5,6‡], **Erzsébet Horváth-Puhó**[5]

**1** Clinical Excellence Research Center, School of Medicine, Stanford University, Stanford, California, United States of America, **2** Department of Pediatrics, The Hospital for Sick Children, University of Toronto, Toronto, Ontario, Canada, **3** Institute of Health Policy, Management, and Evaluation, University of Toronto, Toronto, Ontario, Canada, **4** School of Medicine, New York University, New York, New York, United States of America, **5** Department of Clinical Epidemiology, Aarhus University Hospital, Aarhus, Denmark, **6** Division of Epidemiology, Department of Health Research and Policy, Stanford University, Stanford, California, United States of America

he These authors contributed equally to this work.
‡ These authors are joint senior authors on this work.
* nirav.shah@stanford.edu

**Data Availability Statement:** Danish law does not allow researchers to share raw data from the registries with third parties. Data can be accessed

## Abstract

### Objective

This population-based, matched cohort study aimed to evaluate utilization of health care services by mothers of children with major congenital anomalies (MCAs), compared to mothers of children without MCAs over a 20-year post-birth time horizon in Denmark.

### Methods

Our analytic sample included mothers who gave birth to an infant with a MCA (n = 23,927) and a cohort of mothers matched to them by maternal age, parity and infant's year of birth (n = 239,076). Primary outcomes were period prevalence and mothers' quantity of health care utilization (primary, inpatient, outpatient, surgical, and psychiatric services) stratified by their child's age (*i.e.*, ages 0–6 = before school, ages 7–13 = pre-school + primary education, and ages 14–18 = secondary education or higher). The secondary outcome measure was length of hospital stays. Outcome measures were adjusted for maternal age at delivery, parity, marital status, income quartile, level of education in the year prior to the index birth, previous spontaneous abortions, maternal pregnancy complications, maternal diabetes, hypertension, alcohol-related diseases, and maternal smoking.

### Results

In both cohorts the majority of mothers were between 26 and 35 years of age, married, and employed, and 47% were primiparous. Mothers of infants with anomalies had greater utilization of outpatient, inpatient, surgical, and psychiatric services, compared with mothers in the

by researchers through application to the Danish Data Protection Agency(dt@datatilsynet.dk) and the Danish Health Data Authority (kontakt@sundhedsdata.dk).

**Funding:** The authors received no specific funding for this work.

**Competing interests:** The authors have declared that no competing interests exist.

matched cohort. Inpatient service utilization was greater in the exposed cohort up to 13 years after a child's birth, with the highest risk in the first six years after birth [adjusted risk ratio, 1.13; 95% confidence interval (CI), 1.12–1.14], with a decrease over time. Regarding the quantity of health care utilization, the greatest difference between the two groups was in inpatient service utilization, with a 39% increased rate in the exposed cohort during the first six years after birth (adjusted rate ratio, 1.39; 95% CI, 1.37–1.42). During the first 6 years after birth, mothers of children with anomalies stayed a median of 6 days (interquartile range [IQR], 3–13) in hospital overall, while the comparison cohort stayed a median of 4 days (IQR, 2–7) in hospital overall. Rates of utilization of outpatient clinics (adjusted rate ratio, 1.36; 95% CI, 1.29–1.42), as well as inpatient (adjusted rate ratio, 1.77; 95% CI, 1.68–1.87), and surgical services (adjusted rate ratio, 1.33; 95% CI, 1.26–1.41) was higher in mothers of children with multiple-organ MCAs during 0 to 6 years after birth. Among mothers at the lowest income levels, utilization of psychiatric clinic services increased to 59% and when their child was 7 to 13 years of age (adjusted rate ratio, 1.59; 95% CI, 1.24–2.03).

## Conclusion

Mothers of infants with a major congenital anomaly had greater health care utilization across services. Health care utilization decreased over time or remained stable for outpatient, inpatient, and surgical care services, whereas psychiatric utilization increased for up to 13 years after an affected child's birth. Healthcare utilization was significantly elevated among mothers of children with multiple MCAs and among those at the lowest income levels.

## Introduction

Major congenital anomalies (MCAs) affect 2% to 3% of newborns in the United States and Europe [1–3]. With advancements in medical technology and intensive care, a majority of these infants now survive into adulthood [4]. Yet 71%-96% of children with MCAs continue to experience complex chronic conditions and disabilities [4]. Many require ongoing complex care, which poses significant physical, psychological, social, and financial burdens on parents, particularly mothers [5]. Mothers of children with an MCA are at a higher risk of cardiovascular disease [6], mental disorders (*e.g.*, psychological distress, anxiety, and/or depression) [7–9], and mortality [10]. These health issues may increase rates of health care utilization, but little is known about this association. Understanding the utilization patterns of the caregiving mothers will promote evidence-informed planning, design, and delivery of appropriate integrated care for children with MCAs and their mothers.

The objectives of this study were to compare health care utilization among mothers of children with MCAs to that of mothers of children without MCAs. We hypothesized that mothers of affected children would have greater health care utilization across multiple categories of utilization.

## Methods

### Study design, setting, and data

This was a population-based matched cohort study. Denmark has a free tax supported health care system [11]. The Danish Medical Birth Registry and Danish Civil Registration System

were used to establish the study cohorts. We linked three individual-level national registries—the Danish National Patient Registry, the Danish Psychiatric Central Research Register, and the Danish National Health Service Register—using the unique personal identifier assigned to all residents of Denmark at birth or upon immigration. A detailed description of the registries is provided elsewhere (eAppendix 1 in S1 File) [6, 11, 12].

The study was reported to the Danish Data Protection Agency. Informed consent was not required according to Danish law, as the study comprised secondary analysis of registry data identified prior to the analysis. The study is reported according to the Strengthening the Reporting of Observational Studies in Epidemiology (STROBE) guidelines (eAppendix 2 in S1 File).

## Study cohorts

Our analytic sample included 23,927 women who gave live birth to a singleton with an MCA between January 1, 1997, and December 31, 2017, as recorded in the Danish Medical Birth Registry. These women were followed until December 31, 2018, emigration, loss to follow-up, or death, whichever occurred first. We used ICD-10 diagnosis codes from the Danish National Patient Registry to identify infants with an MCA according to the European Surveillance of Congenital Anomalies Classification System (eAppendix 3 in S1 File). A second cohort of 239,076 women was randomly matched for comparison purposes. Matching was done with replacement [13] and 15.5% of all women from the comparison cohort were used more than once. Women in the comparison cohort were matched up to 10:1 to each affected mother by maternal age at delivery, parity (1, 2, and ≥ 3 children), and infant's year of birth. We excluded women lacking a minimum of two years of data prior to giving birth, which was needed to ascertain pre-existing diagnoses. Mothers with missing values for matching variables were also excluded.

## Study outcomes

The primary outcome of interest was health care utilization stratified by child age (*i.e.*, ages 0–6 = before school, ages 7–13 = preschool + primary education, ages 14–18 = secondary education or higher), based on age cutoffs used in the Danish educational system. We measured utilization of each primary, inpatient, outpatient clinic, surgical, and psychiatric health service. Denmark provides universal health care and primary care, including consultation visits with general practitioners and midwives as well as ultrasound scans [14], for all mothers. We do not present total health care utilization across all settings because this is weighed heavily toward primary care, and thus does not effectively discern among the potentially different needs and patterns of care in the different settings. Health care utilization of each primary, inpatient, outpatient, surgical, and psychiatric health service was defined both as the period prevalence and as the quantity of each service. Primary care services included consultations with general practitioners (including routine, telephone, email, and after-hours) and private practicing medical specialist contacts, almost all of whom have contracts with the National Health Service (additional information on the list of medical specialists included in our analyses is provided in eAppendix 3 in S1 File). Inpatient care included all hospital admissions registered in the Danish National Patient Registry after the date of delivery. Outpatient clinic visits included all hospital-based ambulatory clinic visits, excluding emergency room visits. Surgical care included all surgical procedures in the hospitals after the date of child's birth, except for procedures on male genital organs. When multiple surgical procedures were reported on the same day, we counted them separately. Psychiatric service use was measured separately for clinic visits and inpatient care using data from the Danish Psychiatric Central Research Register (eAppendix 3

in S1 File). Finally, length of hospital stay was defined as days in a hospital (excluding the admission days related to giving the birth) and analyzed as a separate outcome variable.

## Covariates

The following covariates were defined: maternal age at delivery, marital status, income quartile, level of education as of the year prior to giving live birth ("index birth"), parity, previous spontaneous abortions, and maternal index pregnancy complications [*e.g.*, placental complications (including pre-eclampsia, gestational hypertension, and placental abruption/infarction); non-placental complications (including intrauterine hypoxia/birth asphyxia, uterine rupture, umbilical cord prolapse, vasa previa, amniotic fluid embolism, and fetal hemorrhage); diabetes; and alcohol-related diseases]. For mothers with more than one live birth of an infant with MCA, the first affected pregnancy was assigned as the index pregnancy. In addition, we extracted data on chronic hypertension, diabetes, and alcohol use prior to the index date, and on smoking history during the index pregnancy, as each is associated independently with elevated risk of congenital anomalies and increased use of health care services [15–17]. Other maternal medical history was summarized using a modified Charlson comorbidity index score that excluded hypertension, diabetes, and alcohol use. As information on body mass index (BMI) became available after the beginning of the study period (2004), BMI data was used only in a sensitivity analysis.

## Statistical analysis

We calculated the period prevalence of health care utilization among mothers who gave birth to infants with MCAs and among the mothers of infants without MCAs, stratified by the child age categories described above. We used multivariable modified Poisson regression models with robust variance estimators to estimate unadjusted and adjusted risk ratios, accounting for repeated measures in the same individuals [16]. The adjusted model included all covariates listed above. Next, we examined the number of mothers using each health care service annually, from two years before their child's birth until the end of follow-up. The quantity of service utilization by follow-up time, reflecting the child's age category, then was calculated and analyzed using a negative binomial regression model with a robust variance estimator to account for overdispersion. The logarithm of follow-up time was used as an offset parameter. Crude and adjusted rate ratios were estimated with 95% confidence intervals (CIs). We analyzed the length of hospital stays using a generalized linear regression model with lognormal distribution. Then we performed stratified analyses, comparing the cohort of mothers of children affected by MCA with the comparison cohort according to maternal age at index delivery ($\leq 25$ years, 26–35 years, $> 35$ years), the year of delivery (1997–2003, 2004–2010, 2011–2017), infant prematurity ($< 37$ weeks and $\geq 37$ weeks), and infant death during the first year of life. Next, we evaluated the modified effect of bearing children with MCAs according to socioeconomic status (i.e., income, education, immigration) and MCA-related history (*i.e.*, number of pregnancies with MCAs).

We also conducted stratified analyses to investigate whether other important risk factors, such as mothers' health-related behaviors (*i.e.*, alcohol use, smoking history) and mother's own health conditions, were associated with increased health care utilization. To assess a potential dose-response association between the exposure and the outcome, we stratified the main model by the number of child hospitalizations within the first year following birth (*i.e.*, 0, 1–3, 4) and by affected organ system (*i.e.*, a single organ vs. multiple organs). We also conducted sensitivity analyses to assess the robustness of our results. This included alternative model specification, with additional adjustment for BMI. Finally, we repeated the main

analyses excluding mothers with previous psychiatric disorders that might have been associated with risk of accessing psychiatric health care. All data analyses were performed between October 1, 2020, and April 28, 2021, using SAS 9.4 (SAS Institute, Cary, NC).

## Results

### Characteristics of mothers of children with MCAs and mothers in the comparison cohort

Among all 1,273,917 eligible mothers, 23,927 (1.9%) births with MCAs were identified (Fig 1). Table 1 shows that the characteristics of mothers in both cohorts were the same in terms of demographics (age at delivery, marital status, education, income, mortality). The majority were married and employed, and median age at index delivery was 30 years (interquartile range [IQR], 27–34 years) in both cohorts. Approximately 47% of all mothers were primiparous. Compared to mothers of children without MCAs, mothers in the MCA cohort had

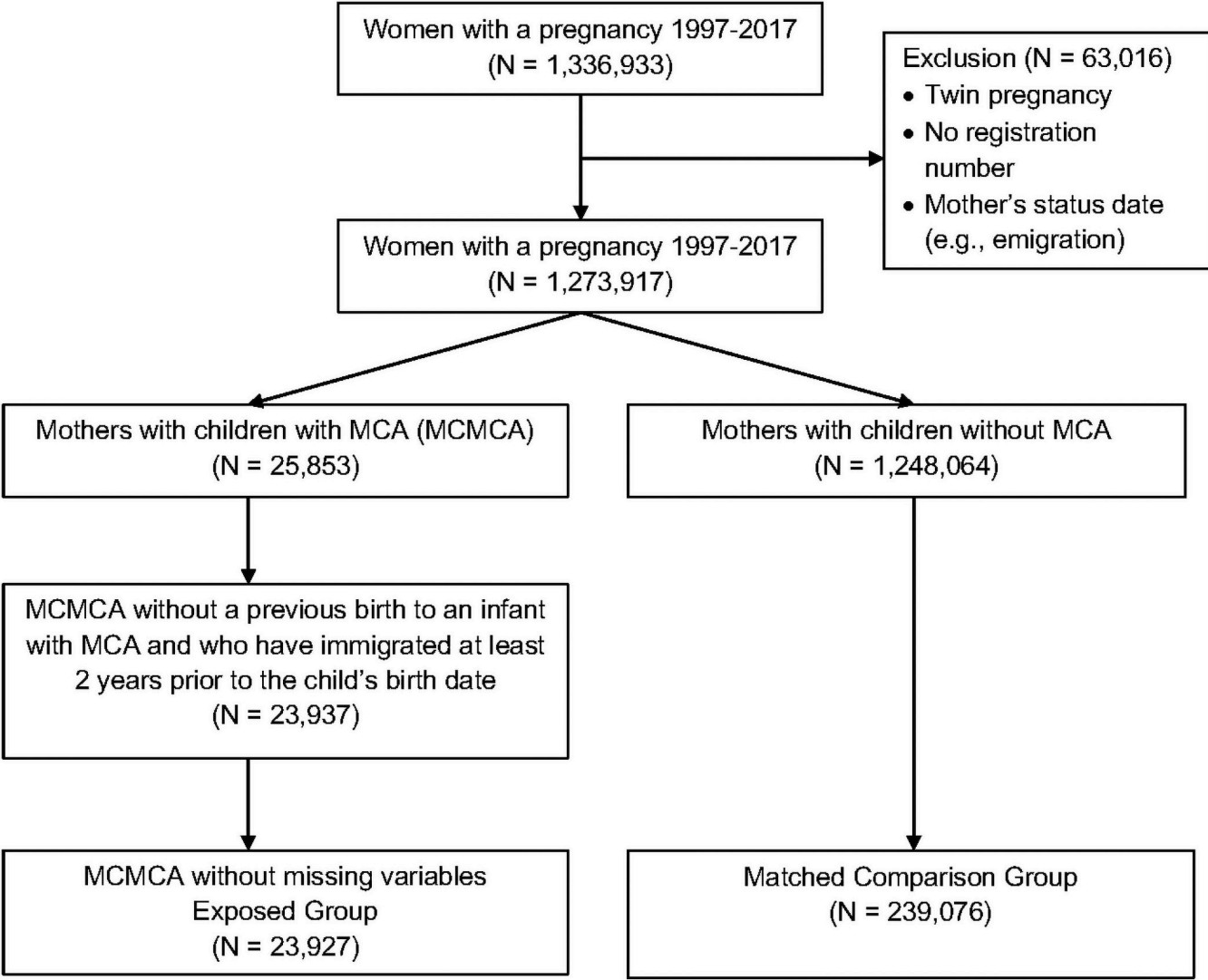

**Fig 1. Flow diagram of cohort derivation.**

**Table 1. Characteristics of mothers included in the cohorts and their infants.**

| Characteristics | Mothers with a child without major congenital anomalies No. (%) | Mothers with a child with major congenital anomalies No. (%) |
|---|---|---|
| Mothers | 239076 | 23927 |
| Age at index delivery | | |
| ≤25 years | 33868 (14.2) | 3378 (14.1) |
| 26–35 | 157702 (66.0) | 15776 (65.9) |
| 36+ | 47506 (19.9) | 4773 (19.9) |
| Parity | | |
| 1 | 112939 (47.2) | 11300 (47.2) |
| 2 | 81584 (34.1) | 8163 (34.1) |
| ≥3 | 44553 (18.6) | 4464 (18.7) |
| Marital status | | |
| Married/registered partnership | 129017 (54.0) | 12753 (53.3) |
| Single/Divorced/Widowed/ Living alone | 104426 (43.7) | 11144 (46.6) |
| Income quartile | | |
| ≤25% | 59867 (25.0) | 6228 (26.0) |
| 26–50% | 59634 (25.0) | 6239 (26.1) |
| 51–75% | 59537 (24.9) | 5890 (24.6) |
| 76–100% | 59780 (25.0) | 5538 (23.1) |
| Highest attained education | | |
| Primary or secondary | 78067 (32.7) | 8305 (34.7) |
| Vocational or post-secondary | 76179 (31.9) | 7727 (32.3) |
| Bachelor or higher | 75745 (31.7) | 6963 (29.1) |
| Employment status | | |
| Employed | 184034 (77.0) | 17786 (74.3) |
| Outside workforce | 39027 (16.3) | 4420 (18.5) |
| Unemployed | 14078 (5.9) | 1472 (6.2) |
| Pregnancy history | | |
| Spontaneous abortions | 36258 (15.2) | 4026 (16.8) |
| Stillbirths | 22 (0) | 10 (0) |
| Index pregnancy complications | | |
| Placental | 11213 (4.7) | 1681 (7.0) |
| Non Placental | 952 (0.4) | 195 (0.8) |
| Multiple pregnancies with MCAs during follow-up | | |
| Single MCA pregnancy | - | 23384 (97.7) |
| Multiple MCA pregnancies | - | 543 (2.3) |
| Medical history | | |
| Diabetes | 1353 (0.6) | 329 (1.4) |
| Chronic hypertension | 1404 (0.6) | 209 (0.9) |
| Alcohol-related diseases | 3284 (1.4) | 423 (1.8) |
| Psychiatric medical history | | |
| Pre-birth any mental health diagnoses | 12154 (5.1) | 1468 (6.1) |
| Any mental health diagnoses during follow up | 13288 (5.6) | 1727 (7.2) |
| Pre-birth depression diagnoses | 1798 (0.8) | 257(1.1) |
| Depression diagnoses during follow up | 3119 (1.3) | 399 (1.7) |

(*Continued*)

**Table 1.** (Continued)

| Characteristics | Mothers with a child without major congenital anomalies No. (%) | Mothers with a child with major congenital anomalies No. (%) |
|---|---|---|
| Modified Charlson Comorbidity Index Score | | |
| 0 | 224246 (93.8) | 22197 (92.8) |
| 1 | 11940 (5.0) | 1326 (5.5) |
| ≥2 | 2890 (1.2) | 404 (1.7) |
| Maternal Smoking | | |
| No | 184645 (77.2) | 17828 (74.5) |
| Yes | 35232 (14.7) | 3817 (16.0) |
| Maternal BMI (2004–2017) | | |
| <25 | 103183 (64.4) | 9617 (59.9) |
| 25–29 | 32501 (20.3) | 3396 (21.2) |
| ≥30 | 19321 (12.1) | 2393 (14.9) |
| Duration of follow-up, median (IQR), years | 11.2 (5.9–16.5) | 11.3 (5.9–16.5) |
| Infant | | |
| Sex | | |
| Male | 122107 (51.1) | 14126 (59.0) |
| Female | 115999 (48.5) | 8682 (36.3) |
| Birthweight, g | | |
| ≤2500 | 8757 (3.7) | 3674 (15.4) |
| >2500–4000 | 188233 (78.7) | 16915 (70.7) |
| >4000 | 41412 (17.3) | 3215 (13.4) |
| Gestational age | | |
| 15–36 | 11538 (4.8) | 3965 (16.6) |
| 37–41 | 213120 (89.1) | 18739 (78.3) |
| 42–50 | 13209 (5.5) | 1082 (4.5) |
| Apgar score <7 at 5min | 3463 (1.4) | 1292 (5.4) |
| Year of infant's birth | | |
| 1997–2003 | 78784 (33.0) | 7884 (33.0) |
| 2004–2010 | 79600 (33.3) | 7966 (33.3) |
| 2011–2017 | 80692 (33.8) | 8077 (33.8) |
| Severity of MCA | | |
| Single organ MCA | - | 21354 (89.2) |
| Multiple organ MCAs | - | 2573 (10.8) |
| Number of hospitalizations within the first year | | |
| 0 hospitalization | 185765 (77.7) | 6395 (26.7) |
| 1–3 hospitalization | 50889 (21.3) | 12689 (53.0) |
| ≥4 hospitalization | 2422 (1.0) | 4843 (20.2) |
| Infant death by 1 year | 61 (0.0) | 159 (0.7) |
| Single organ MCA | - | 101 (0.5) |
| Multiple organ MCA | - | 58 (2.3) |
| 0 hospitalization | 23 (0.01) | 31 (0.5) |
| 1–3 hospitalizations | 29 (0.1) | 67 (0.5) |
| ≥4 hospitalizations | 9 (0.4) | 61 (1.3) |
| Mortality with single organ MCA | | |
| by 1 year | - | 101 (0.5) |

(*Continued*)

**Table 1.** (Continued)

| Characteristics | Mothers with a child without major congenital anomalies No. (%) | Mothers with a child with major congenital anomalies No. (%) |
|---|---|---|
| at 0–6 years | - | 223 (1.0) |
| at 7–13 years | - | 26 (0.1) |
| at 14–18 years | - | 16 (0.1) |
| Mortality with multiple organ MCA | | |
| by 1 year | - | 58 (2.3) |
| at 0–6 years | - | 127 (4.9) |
| at 7–13 years | - | 17 (0.7) |
| at 14–18 years | - | 5 (0.2) |
| Mortality with 0 hospitalization within the first year | | |
| by 1 year | 23 (0.01) | 31 (0.5) |
| at 0–6 years | 104 (0.1) | 42 (0.7) |
| at 7–13 years | 33 (0.02) | <5 |
| at 14–18 years | 17 (0.01) | <5 |
| Mortality with 1–3 hospitalizations within the first year | | |
| by 1 year | 29 (0.1) | 67 (0.5) |
| at 0–6 years | 64 (0.1) | 126 (1.0) |
| at 7–13 years | 12 (0.02) | 14 (0.1) |
| at 14–18 years | 13 (0.03) | 9 (0.1) |
| Mortality with >3 hospitalizations within the first year | | |
| by 1 year | 9 (0.4) | 61 (1.3) |
| at 0–6 years | 30 (1.2) | 182 (3.8) |
| at 7–13 years | <5 | 29 (0.6) |
| at 14–18 years | <5 | 9 (0.2) |

slightly more pregnancy-related complications (spontaneous abortions, 16.8% v. 15.2%), comorbidities (diabetes, 1.4% vs. 0.6%), current smoking (16.0% v. 14.7%), and pre-birth mental health diagnosis histories (6.1% vs. 5.1%).

Compared to infants without MCAs, infants with MCAs were more likely to have low birth weights (15.4% vs. 3.7%), prematurity (16.6% vs. 4.8%), low five-minute Apgar scores (5.4% vs. 1.4%) and greater mortality at 1 year (0.7% vs. 0%). Mortality during the first year was 0.5%, 0.5%, and 1.3%, respectively, in children with MCAs, with no hospitalization, 1–3 hospitalizations, and more than 3 hospitalizations, compared to 0.01%, 0.1%, and 0.4% in children without MCAs. Children with multi-organ MCAs had higher mortality than those with single-organ MCAs during the first year (2.3% vs. 0.5%).

## Association between bearing children with MCAs and the period prevalence of health care utilization

The period prevalence of the utilization of outpatient, inpatient, surgical, and psychiatric services was greater in the exposed cohort. In contrast, the period prevalence of the utilization of primary care and the sum across settings did not differ between the exposed and comparison cohorts (Fig 2). Inpatient service utilization was greater in the exposed cohort up to 13 years after a child's birth, with the greatest risk ratio occurring during the first six years after birth

| Period Prevalence of Services | Exposed N (%) | Comparison N (%) | | Unadjusted Risk Ratio (95% CI) | Adjusted Risk Ratio (95% CI) |
|---|---|---|---|---|---|
| **Primary Care** | | | | | |
| 0-6 years after birth | 17797 (100) | 177672 (100) | | 1.00 [1.00 ; 1.00] | 1.00 [1.00 ; 1.00] |
| 7-13 years after birth | 9932 (99.9) | 99227 (99.9) | | 1.00 [1.00 ; 1.00] | 1.00 [1.00 ; 1.00] |
| 14-18 years after birth | 4334 (99.8) | 43343 (99.8) | | 1.00 [1.00 ; 1.00] | 1.00 [1.00 ; 1.00] |
| **Outpatient Clinic** | | | | | |
| 0-6 years after birth | 16050 (90.2) | 157403 (88.6) | | 1.02 [1.01 ; 1.02] | 1.02 [1.01 ; 1.02] |
| 7-13 years after birth | 8438 (84.9) | 83071 (83.6) | | 1.02 [1.01 ; 1.02] | 1.01 [1.00 ; 1.02] |
| 14-18 years after birth | 3655 (84.2) | 35497 (81.7) | | 1.03 [1.02 ; 1.04] | 1.03 [1.01 ; 1.04] |
| **Psychiatric Clinic** | | | | | |
| 0-6 years after birth | 1030 (5.8) | 8669 (4.9) | | 1.19 [1.11 ; 1.26] | 1.14 [1.07 ; 1.21] |
| 7-13 years after birth | 668 (6.7) | 5790 (5.8) | | 1.15 [1.07 ; 1.25] | 1.12 [1.03 ; 1.21] |
| 14-18 years after birth | 233 (5.4) | 2068 (4.8) | | 1.13 [0.99 ; 1.29] | 1.09 [0.96 ; 1.24] |
| **Psychiatric Inpatient** | | | | | |
| 0-6 years after birth | 273 (1.5) | 2454 (1.4) | | 1.11 [0.98 ; 1.26] | 1.05 [0.92 ; 1.19] |
| 7-13 years after birth | 227 (2.3) | 1793 (1.8) | | 1.27 [1.10 ; 1.45] | 1.21 [1.05 ; 1.38] |
| 14-18 years after birth | 71 (1.6) | 587 (1.4) | | 1.21 [0.95 ; 1.55] | Inestimable |
| **Inpatient Care** | | | | | |
| 0-6 years after birth | 13372 (75.1) | 117805 (66.3) | | 1.13 [1.12 ; 1.14] | 1.13 [1.12 ; 1.14] |
| 7-13 years after birth | 4999 (50.3) | 45821 (46.1) | | 1.09 [1.07 ; 1.11] | 1.07 [1.05 ; 1.10] |
| 14-18 years after birth | 1486 (34.2) | 13344 (30.7) | | 1.11 [1.07 ; 1.16] | 1.09 [1.04 ; 1.14] |
| **Surgical Care** | | | | | |
| 0-6 years after birth | 11612 (65.2) | 108645 (61.1) | | 1.07 [1.05 ; 1.08] | 1.06 [1.05 ; 1.07] |
| 7-13 years after birth | 5464 (55.0) | 51930 (52.3) | | 1.05 [1.03 ; 1.07] | 1.04 [1.02 ; 1.06] |
| 14-18 years after birth | 1966 (45.3) | 18501 (42.6) | | 1.06 [1.03 ; 1.10] | 1.05 [1.02 ; 1.09] |

**Fig 2. Association of the mothers of children with MCAs and healthcare utilization period prevalence.**

(adjusted risk ratio, 1.13; 95% CI, 1.12–1.14), and then decreasing over time. Surgical service utilization showed a similar pattern, with most utilization occurring during the first six years after birth (adjusted risk ratio, 1.06; 95% CI, 1.05–1.07). Thereafter, surgical utilization slightly decreased but remained higher than in the comparison cohort. Psychiatric service utilization also was greater in the exposed cohort, with a more sustained pattern of utilization compared with the other types of services. Mothers of children with MCAs used more psychiatric clinic services compared to the comparison cohort both during the first six years after birth (adjusted risk ratio, 1.14; 95% CI, 1.07–1.21), and when their children were 7–13 years of age (adjusted risk ratio, 1.12; 95% CI, 1.03–1.21). Psychiatric inpatient service use was similar between the exposed and comparison cohorts when children were 0–6 years of age (adjusted risk ratio, 1.05; 95% CI, 0.92–1.19), but increased in the exposed cohort when the child was 7–13 years of age (adjusted risk ratio, 1.21; 95% CI, 1.05–1.38).

## Association between bearing children with MCAs and quantity of health care utilized

Unadjusted annual mean health care service utilization is presented in Figs 3–8. The exposed cohort utilized slightly more health care services overall, except for psychiatric inpatient services. This pattern remained after we adjusted for the covariates listed above (Fig 9). In general, the quantity of health care utilized was greater in the MCA cohort during the first six years

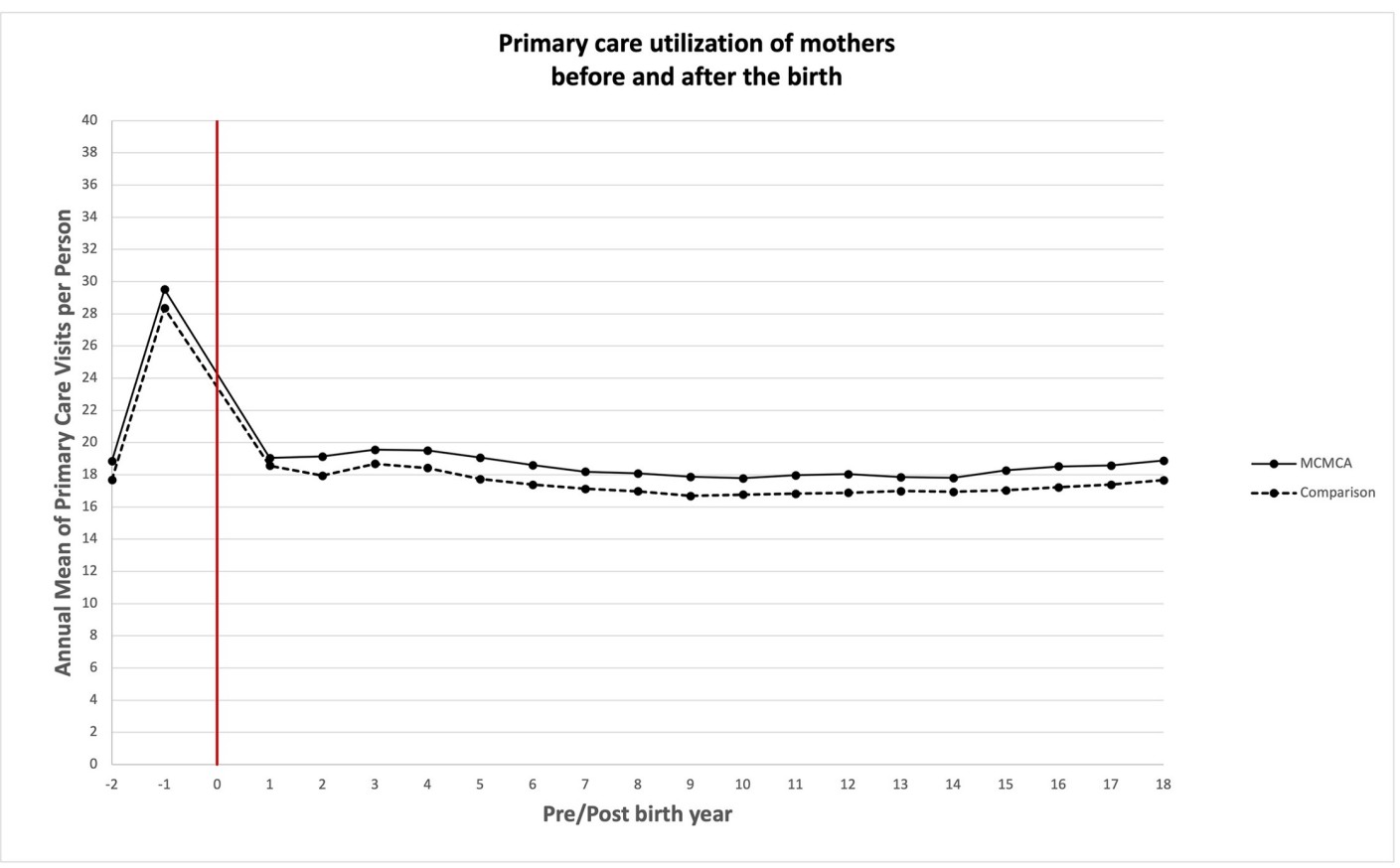

**Fig 3. Unadjusted annual mean primary care utilization before and after the birth during the follow up period.**

after birth, and then became similar to that for comparison group over time. Inpatient service utilization was greater in the exposed cohort across all follow-up periods, with utilization increasing by 39% (adjusted rate ratio, 1.39; 95% CI, 1.37–1.42), 14% (adjusted rate ratio, 1.14; 95% CI, 1.10–1.19), and 13% (adjusted rate ratio, 1.13; 95% CI, 1.05–1.22) during years 0–6, 7–13, and 14–18 of the follow-up period, respectively. Outpatient and surgical service utilization also showed a similar pattern of increased utilization occurring over the entire follow-up period.

Psychiatric clinic service utilization was greater among mothers in the exposed cohort compared to mothers in the comparison cohort when their child was 7 to 13 years of age. Psychiatric clinic utilization did not differ between the two groups during the initial six years after birth (adjusted rate ratio, 1.04; 95% CI, 0.90–1.21), but when the child was 7 to 13 years of age, it was 22% greater in the exposed cohort (adjusted rate ratio, 1.22; 95% CI, 1.02–1.45).

## Association between bearing children with MCA and lengths of hospital stay

Mothers of children with MCA had longer hospital stays. In the first 6 years after birth, mothers of children with MCA stayed a median of 6 days (IQR, 3–13) in hospital overall, while mothers in the comparison cohort stayed a median of 4 days (IQR, 2–7) overall.

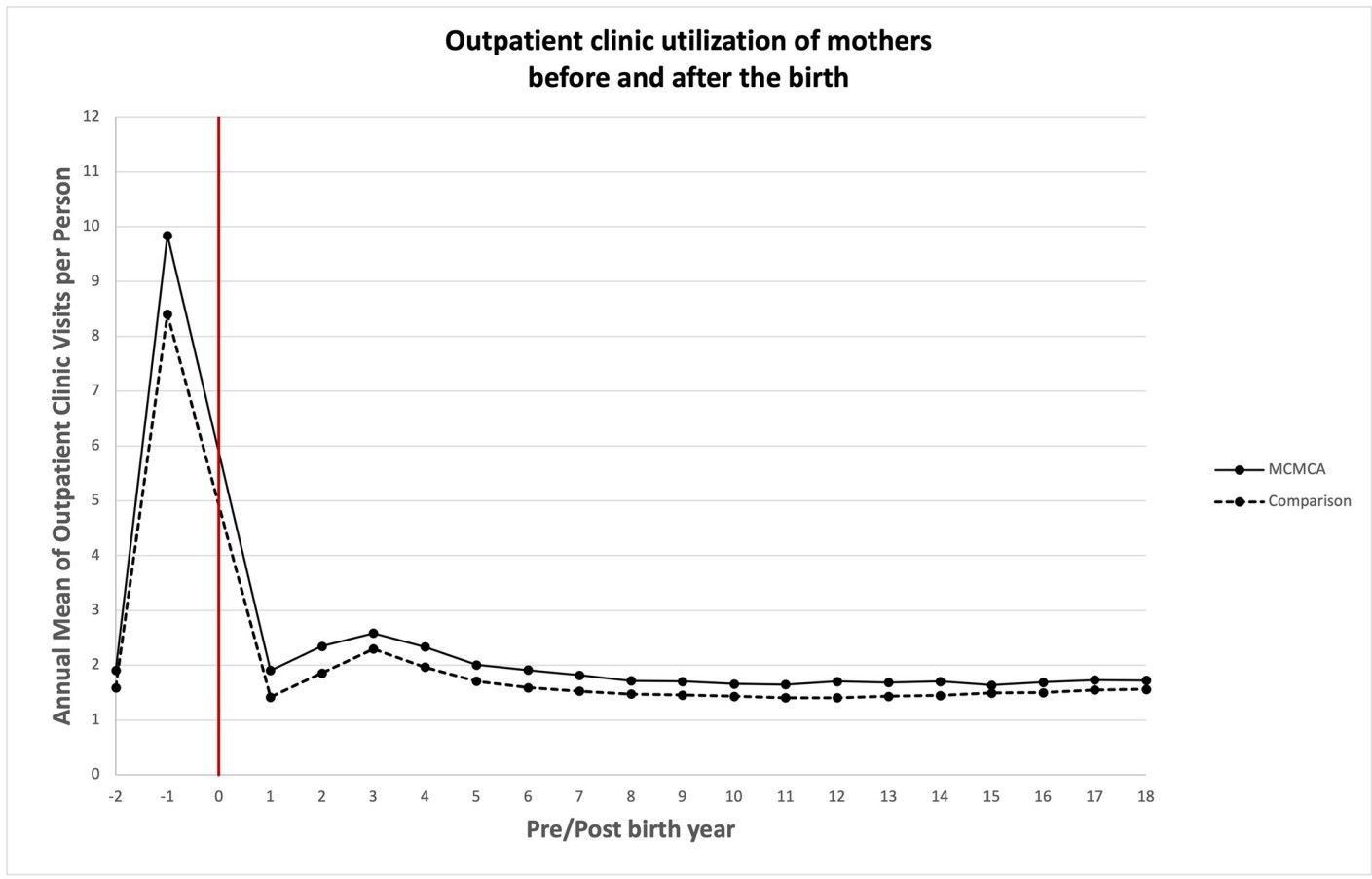

**Fig 4. Unadjusted annual mean outpatient clinic utilization before and after the birth during the follow up period.**

## Stratified analyses

Mothers who bore children with multi-organ MCAs stayed in the hospital about a half a day longer overall than mothers whose children had single-organ MCAs during the 0 to 6 years after birth (adjusted mean difference, 2.00 (95% CI, 1.90–2.10) for multi-organ MCAs vs. 1.49 (95% CI, 1.46–1.52) for single-organ MCAs).

Mothers of children with multiple-organ MCAs also utilized health care to a greater degree than mothers of children with single-organ MCAs, especially in the first six years after birth (eAppendices 5 and 6 in S1 File). During the 0 to 6 years after birth, the rate of utilization of outpatient clinics (adjusted rate ratio, 1.36; 95% CI, 1.29–1.42) as well as inpatient services (adjusted rate ratio, 1.77; 95% CI, 1.68–1.87) and surgical services (adjusted rate ratio, 1.33; 95% CI, 1.26–1.41) was higher in mothers of children with multiple-organ MCAs.

Mothers of children with multiple hospitalizations utilized health care less often (eAppendices 7 and 8 in S1 File). Compared to mothers who bore children without MCAs, mothers of children with an MCA who had no hospitalizations used more outpatient clinic care (adjusted rate ratio, 1.30; 95% CI, 1.26–1.34), while mothers of children with MCAs who had more than 3 hospitalizations used less outpatient clinic care up to 13 years after birth (adjusted rate ratio, 0.85; 95% CI, 0.80–0.90 during the 0 to 6 years after birth; adjusted rate ratio, 0.85; 95% CI, 0.75–0.95 during the 7 to 13 years after birth). The same trends were observed for inpatient admissions and surgical care services.

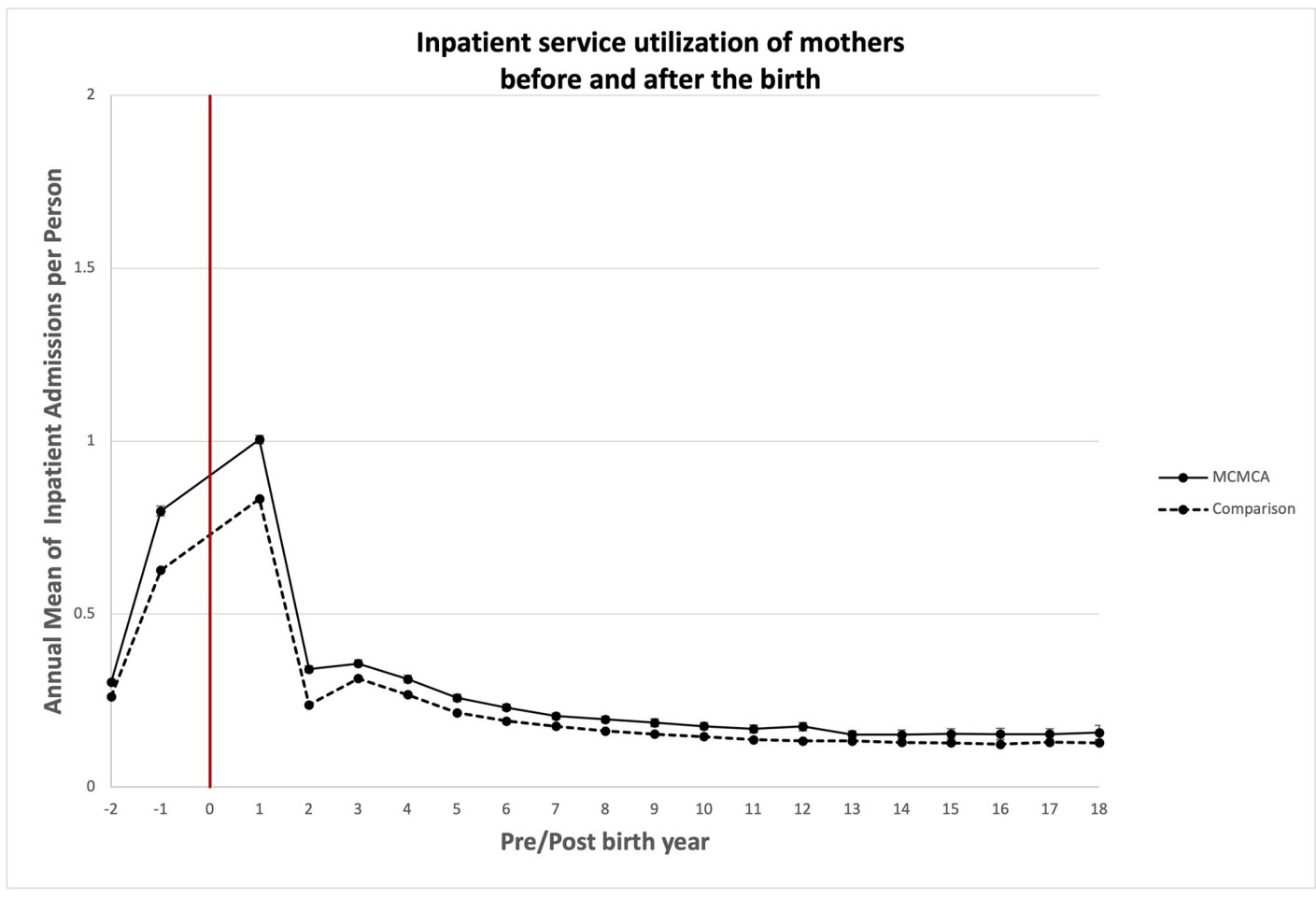

**Fig 5. Unadjusted annual mean inpatient service utilization before and after the birth during the follow up period.**

Utilization of psychiatric clinic services was greater for exposed mothers in the lowest income quartile, which increased to 59% when the child was 7 to 13 years of age (adjusted rate ratio, 1.59; 95% CI, 1.24–2.03; eAppendices 9 and 10 in S1 File). Psychiatric inpatient service utilization was 57% (adjusted rate ratio, 1.57; 95% CI, 1.15–2.15) and 76% (adjusted rate ratio, 1.76; 95% CI, 1.07–2.89) greater in the exposed cohort in the lowest income quartile, when children were 0–6 years and 7–13 years of age, respectively.

Compared to mothers of children with a single MCA pregnancy history, mothers of children with multiple MCA pregnancy histories used outpatient clinic services (adjusted rate ratio, 1.83 vs. adjusted rate ratio 1.18), inpatient services (adjusted rate ratio 1.85 vs. adjusted rate ratio 1.38), and surgical care services (adjusted rate ratio 1.61 vs. adjusted rate ratio 1.14) to a greater extent during the first 6 years after birth (eAppendices 11 and 12 in S1 File). This pattern continued until 13 years after birth. Health care utilization did not differ by other socioeconomic indicators (education and immigration) or mothers' health-related behaviors.

Increased health care utilization associated with mothers of children with MCA were observed for mothers with similar levels of medical conditions. Among mothers with the existing medical conditions (i.e., the Modified Charlson Comorbidity Index $\geq$ 1), the exposed cohort used more outpatient clinic services, inpatients services, and surgical care services compared to the unexposed cohort (eAppendices 13 and 14 in S1 File).

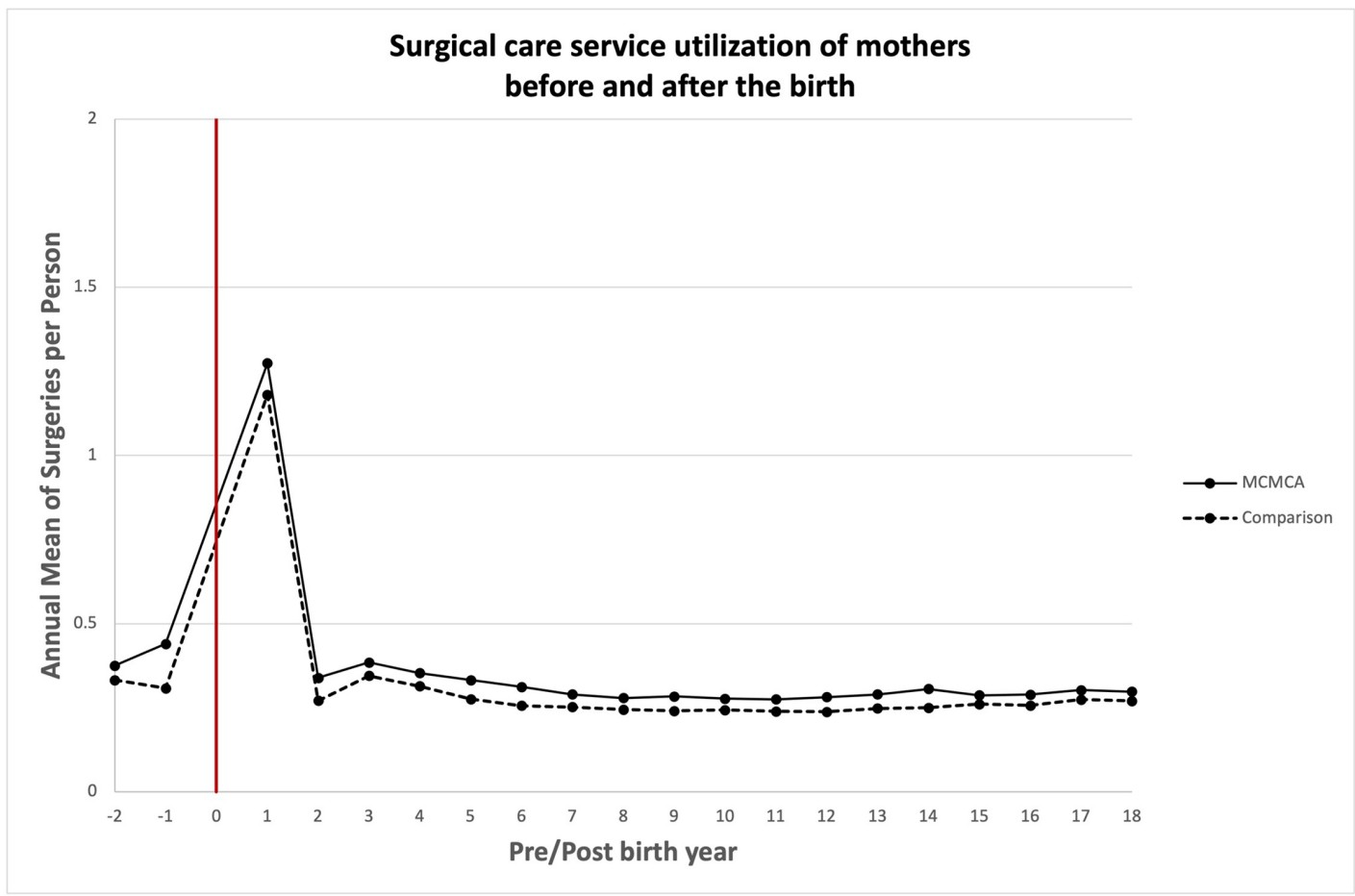

**Fig 6. Unadjusted annual mean surgical care service utilization before and after the birth during the follow up period.**

## Sensitivity analyses

Sensitivity analyses showed results consistent with the main analyses across the study outcomes, indicating greater likelihood of health care utilization for mothers of children with MCAs. We estimated period prevalence and the quantity of health care utilization from the models adjusted for BMI as well as other covariates in the 2004–2017 period. The results were nearly identical to those from the models that did not adjust for BMI in the entire follow-up period. When we excluded mothers with a history of psychiatric disorders or depression to address the potential differences in psychiatric health risks, the results also were consistent with those for the main analyses.

## Discussion

Our study on health care utilization among mothers of infants with a major congenital anomaly showed greater health care utilization across services, compared to a matched cohort of women with unaffected infants over a 20-year post-birth time horizon. The magnitude varied across settings, from small differences in use of outpatient clinics to moderate differences in use of inpatient and psychiatric clinic services. Health care utilization among exposed mothers decreased as their children grew older or remained stable for outpatient, inpatient, and surgical

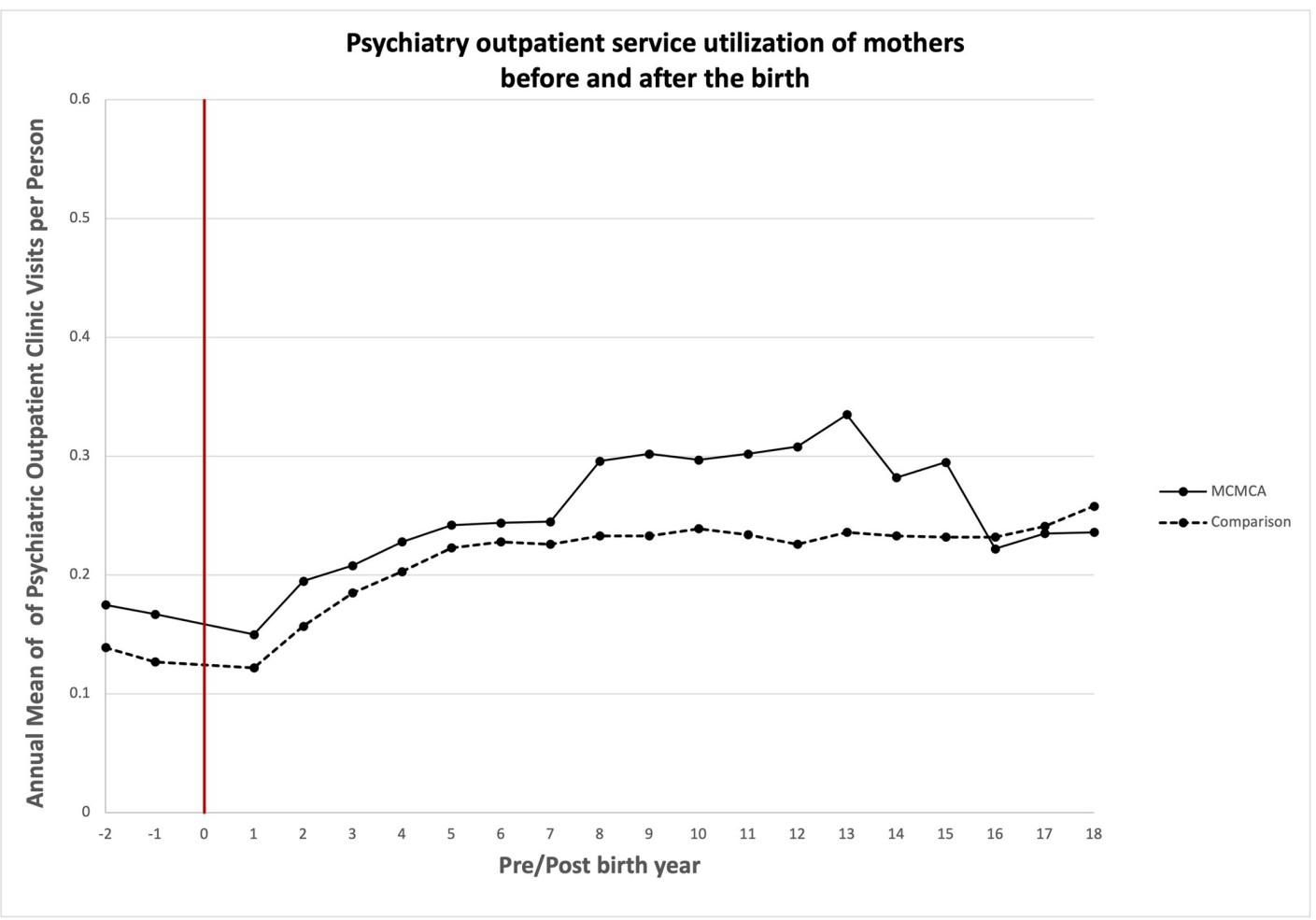

**Fig 7. Unadjusted annual mean psychiatric outpatient service utilization before and after the birth during the follow up period.**

care services, whereas psychiatric utilization increased for up to 13 years after an affected child's birth. We observed significant increases in health care utilization among mothers of children with multiple MCAs, among those at the lowest income levels, and among those with previous mental health issues.

Our results showed that increased maternal health risks were associated with a greater likelihood of health care utilization for mothers of chronically ill children. Our findings suggest that the physical and psychological challenges of caring for a chronically ill child require more health care. Our study is the first to confirm that mothers of children with MCAs are more likely to use health services, ranging from 1% to 39% more, compared to mothers with unaffected children, depending on the services and time since a child's birth. Although the magnitude of increased health care utilization was small to moderate and depended on type of service, the health system impact is important because MCAs are not uncommon. In the analytic sample of approximately 25,000 mothers of children with an MCA, the mothers had an average of 2 times more outpatient clinic visits and 0.5 more inpatient admissions over six years per person, compared to a matched cohort of mothers with unaffected children. This could represent 50,000 more outpatient clinic visits and 12,500 more inpatient stays over six years—equivalent to more than 8,000 outpatient clinic visits and 2,000 hospitalizations per

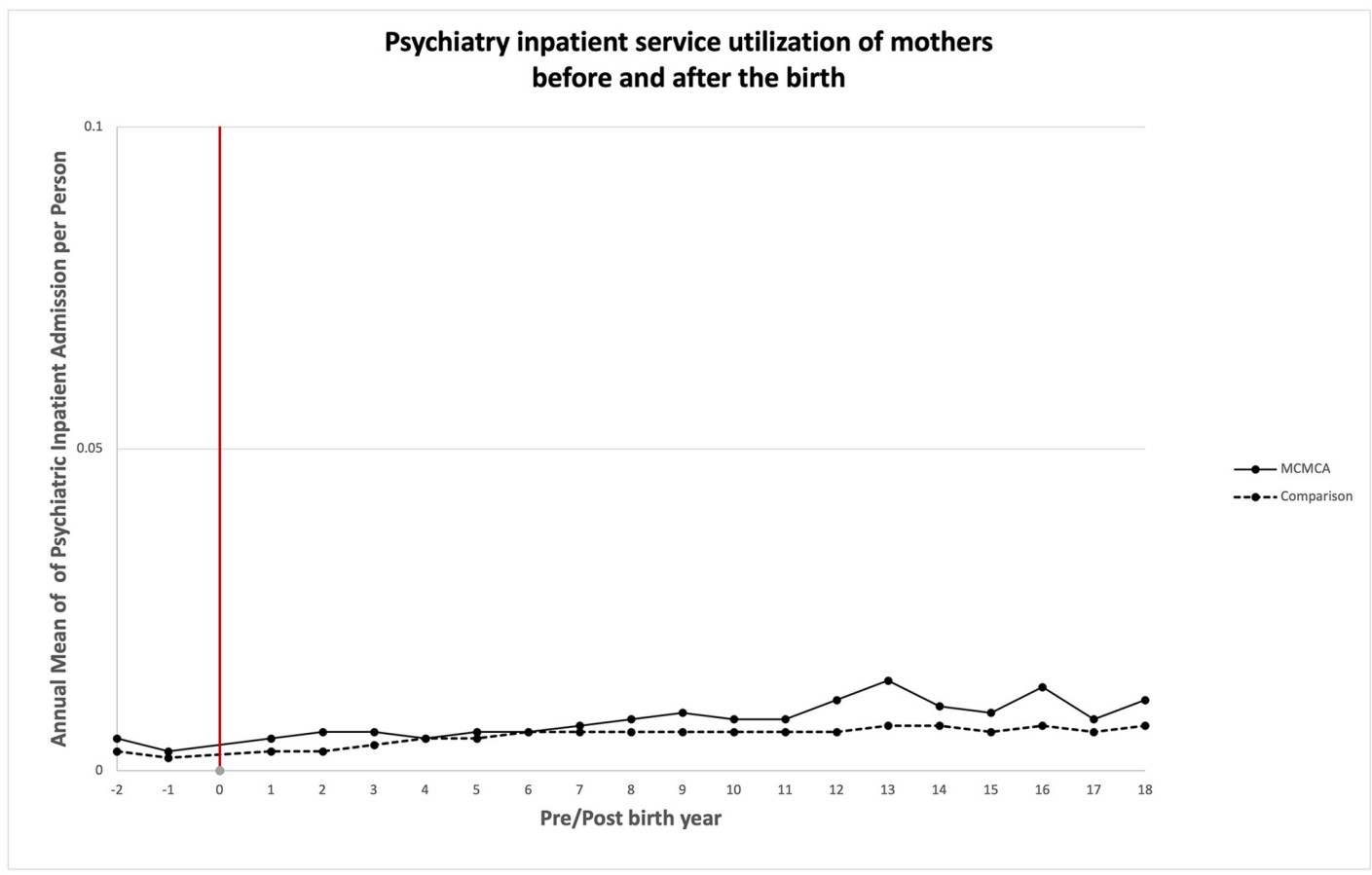

**Fig 8. Unadjusted annual mean psychiatric inpatient service utilization before and after the birth during the follow up period.**

year. Considering that health care spending per capita in Denmark was 5,568 USD in 2019 [17], a 1% increase in health care utilization means an increase of 325 million USD annually (5,568 x 0.01 x 5.840 million population [18] in Denmark). Reducing 1% of health care utilization would be sufficient to cover annual out-of-pocket expenditures [19] for 7% of Danes. High utilization of health care and related health care costs could be greater in the US, where health care spending per capita is more than twice Denmark's and health care system performance is poorer than other countries [20].

To date, research evaluating health care utilization among mothers of chronically ill children has been limited. Our results are consistent with existing evidence for other caregiving populations (*e.g.*, those caring for dementia patients) that also found greater health care utilization among the caregivers [21, 22]. This study was conducted in a longitudinal setting among mothers in Denmark, where the national health care system fully covers primary, specialist, inpatient, and mental health services. This allowed us to examine the long-term effects of caregiver burden while eliminating the effects of access to care or the ability to pay for services [23]. Our results underscore that the long-term burden of caring for chronically ill children has significant effects on multiple dimensions of caregiver health and health care use over time.

Our findings indicate that in general, mothers with chronically ill children had significantly greater health care utilization than unexposed mothers, with the magnitude of utilization

| Quantity of Services | Exposed<br>Median (IQR) | Comparison<br>Median (IQR) | | Unadjusted Rate Ratio (95% CI) | Adjusted Rate Ratio (95% CI) |
|---|---|---|---|---|---|
| **Primary Care** | | | | | |
| 0-6 years after birth | 100 (68-143) | 95 (66-135) | | 1.06 [1.05 ; 1.07] | 1.04 [1.03 ; 1.05] |
| 7-13 years after birth | 104 (69-157) | 99 (66-148) | | 1.06 [1.05 ; 1.08] | 1.05 [1.03 ; 1.06] |
| 14-18 years after birth | 74 (47-114) | 70 (45-107) | | 1.07 [1.05 ; 1.10] | 1.06 [1.04 ; 1.09] |
| **Outpatient Clinic** | | | | | |
| 0-6 years after birth | 9 (3-17) | 7 (2-14) | | 1.22 [1.20 ; 1.24] | 1.20 [1.17 ; 1.22] |
| 7-13 years after birth | 6 (2-15) | 5 (1-13) | | 1.16 [1.13 ; 1.20] | 1.13 [1.09 ; 1.16] |
| 14-18 years after birth | 4 (1-11) | 4 (1-9) | | 1.14 [1.09 ; 1.19] | 1.11 [1.06 ; 1.16] |
| **Psychiatric Clinic** | | | | | |
| 0-6 years after birth | 0 (0-0) | 0 (0-0) | | 1.14 [1.01 ; 1.28] | 1.04 [0.90 ; 1.21] |
| 7-13 years after birth | 0 (0-0) | 0 (0-0) | | 1.26 [1.09 ; 1.45] | 1.22 [1.02 ; 1.45] |
| 14-18 years after birth | 0 (0-0) | 0 (0-0) | | 1.14 [0.90 ; 1.43] | 1.14 [0.86 ; 1.53] |
| **Psychiatric Inpatient** | | | | | |
| 0-6 years after birth | 0 (0-0) | 0 (0-0) | | 1.26 [1.02 ; 1.56] | 1.11 [0.91 ; 1.36] |
| 7-13 years after birth | 0 (0-0) | 0 (0-0) | | 1.38 [1.06 ; 1.79] | 1.21 [0.93 ; 1.59] |
| 14-18 years after birth | 0 (0-0) | 0 (0-0) | | 1.75 [1.04 ; 2.92] | Inestimable |
| **Inpatient Care** | | | | | |
| 0-6 years after birth | 1 (1-3) | 1 (0-2) | | 1.41 [1.38 ; 1.43] | 1.39 [1.37 ; 1.42] |
| 7-13 years after birth | 1 (0-2) | 0 (0-1) | | 1.19 [1.14 ; 1.23] | 1.14 [1.10 ; 1.19] |
| 14-18 years after birth | 0 (0-1) | 0 (0-1) | | 1.19 [1.10 ; 1.28] | 1.13 [1.05 ; 1.22] |
| **Surgical Care** | | | | | |
| 0-6 years after birth | 1 (0-3) | 1 (0-2) | | 1.17 [1.15 ; 1.20] | 1.15 [1.13 ; 1.18] |
| 7-13 years after birth | 1 (0-3) | 1 (0-2) | | 1.13 [1.10 ; 1.17] | 1.10 [1.07 ; 1.14] |
| 14-18 years after birth | 0 (0-2) | 0 (0-2) | | 1.11 [1.05 ; 1.18] | 1.09 [1.03 ; 1.15] |

**Fig 9. Association of the mothers of children with MCAs and healthcare utilization rates.**

decreasing over time. This suggests that caregiving mothers may adapt to their roles and develop internal coping strategies, thus reducing their distress as time passed [24–26]. However, it is important to note that needs change over time and for different services. Primary and outpatient clinic services were most needed for mothers a year before the birth of a child with an MCA, while inpatient and surgical care services were most needed a year after giving birth. The mothers of children with MCAs used psychiatric clinic services consistently across the follow-up period. Our results highlight the need to provide integrated care for mothers of children with MCAs with various long-term focuses. Developing care plans to provide primary and outpatient care promoting prevention, detection, and adequate treatment during the prenatal care period; inpatient and surgical care allowing treatment for physical needs during early childhood; and psychiatric care supporting mental health needs during middle childhood to adolescence are warranted.

We also found that mothers who experienced more than one pregnancy with offspring affected by an MCA were more likely to use health care services than mothers with a single pregnancy with this outcome. These findings highlight the complexity of variables affecting caregivers' health utilization, and the range of services needed to respond to physical and psychological problems. Although caregiving mothers of children with MCAs might physically adapt to their roles, the demands of caregiving, and associated psychological burdens might not ease over time. Future studies evaluating the factors affecting physical and psychological health are warranted.

Our evaluation of dose–response associations yielded conflicting results. Mothers of children with multiple-organ MCAs utilized health care to a greater degree, while mothers of children with multiple hospitalizations utilized health care less. Multiple-organ MCAs, as a proxy for severe MCAs, indicated that mothers with sicker children were more likely to use health care services for their own health issues. Contrary to our hypothesis, mothers' health care use had an inverse relationship with caring for children with multiple hospitalizations. A possible explanation is that mothers' tendency to utilize less health care for their own health may be influenced by increased childcare responsibilities and constrained resources, such as time and finances. Thus, mothers' health needs might be compromised in this context. Another potential explanation is higher mortality rates of children with MCAs. Children with multiple hospitalizations had higher mortality rates during the first year of life as well as in early childhood beyond infancy. Loss of children might attenuate the caregiving burden over time while bringing intense grief and profound distress.

Caring for a child with congenital anomalies does not impact all mothers equally. Our results suggest that caregiving mothers of chronically ill children at the lowest income levels, as well as those with pre-pregnancy mental health problems, were likely to have greater psychiatric service utilization. As these groups potentially face greater challenges, future research should focus on addressing their needs at an early stage.

## Limitations

This study has several limitations. First, the observed associations between mothers of children with MCAs and health care utilization may be underestimated due to the provision of universal free health care and the broad range of social benefits in Denmark. Given that access to health care and social support have a protective effect for maternal health outcomes, the magnitude of increased healthcare utilization could be greater in the US [10]. Second, the observed associations between mothers of children with MCA and health care utilization might be overestimated because the exposed cohort might present with worse health conditions prior to the index birth. To address this issue, we adjusted for factors independently associated with increased use of health care services. We also performed the stratified analyses by these factors and confirmed the robustness of our main results. Third, we cannot rule out the misclassification of marital status because some mothers might not have registered their cohabitation status. This might lead to a high proportion of mothers with non-married status, and we could not examine the role of cohabitation in our study. However, if rates of misclassification are similar between the exposed and unexposed cohorts, this would not bias the results. Fourth, the observed relationship for mothers of children with MCAs and health care utilization may be susceptible to unobserved confounding. In addition, we cannot rule out confounding from unmeasured factors, such as genetics or lifestyle characteristics, or from factors that vary over time. However, our sensitivity analyses confirmed the robustness of our main results. Fifth, our results may not be generalizable to population in other settings with different demographic characteristics. This study was conducted using a sample from Denmark, which is relatively homogeneous in terms of racial and ethnic diversity. Prior studies have shown that stress processing and perceived caregiving burden differ by race and ethnicity [25, 27, 28]; future studies should take these characteristics into account.

## Conclusion

Mothers of children with major congenital anomalies utilized health care 1% to 39% more than unexposed mothers, depending on type of service over a 20-year post-birth time horizon. Caregiving mothers of a chronically ill child may face greater physical and psychological

challenges, which increase maternal health risks. Understanding how challenges faced by caregivers are related to level of health care utilization would clarify the needs of mothers of chronically ill children and help to guide interventions aimed at reducing burdens on caregiver health over time.

## Supporting information

**S1 File.**
(DOCX)

## Acknowledgments

The authors thank Dr. Phil Moore for his help in translating documents for this study.

## Author Contributions

**Conceptualization:** Nirav R. Shah, Kyung Mi Kim, Venus Wong, Eyal Cohen, Sarah Rosenbaum, Eli M. Cahan, Arnold Milstein, Henrik Toft Sørensen, Erzsébet Horváth-Puhó.

**Data curation:** Eyal Cohen, Erzsébet Horváth-Puhó.

**Formal analysis:** Erzsébet Horváth-Puhó.

**Investigation:** Nirav R. Shah, Eyal Cohen, Arnold Milstein, Henrik Toft Sørensen, Erzsébet Horváth-Puhó.

**Methodology:** Nirav R. Shah, Kyung Mi Kim, Eyal Cohen, Arnold Milstein, Henrik Toft Sørensen, Erzsébet Horváth-Puhó.

**Supervision:** Nirav R. Shah, Eyal Cohen, Arnold Milstein, Henrik Toft Sørensen.

**Validation:** Eyal Cohen, Erzsébet Horváth-Puhó.

**Visualization:** Kyung Mi Kim.

**Writing – original draft:** Nirav R. Shah, Kyung Mi Kim, Venus Wong, Eyal Cohen, Sarah Rosenbaum, Erzsébet Horváth-Puhó.

**Writing – review & editing:** Nirav R. Shah, Kyung Mi Kim, Venus Wong, Eyal Cohen, Arnold Milstein, Henrik Toft Sørensen, Erzsébet Horváth-Puhó.

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
