## [Decision Letter · Decision Letter 0]

6 Sep 2021

PONE-D-21-20906Mothers of children with major congenital anomalies have increased health care utilization over a 20-year post-birth time horizonPLOS ONE

Dear Dr. Shah,

Thank you for submitting your manuscript to PLOS ONE. After careful consideration, we feel that it has merit but does not fully meet PLOS ONE’s publication criteria as it currently stands. Therefore, we invite you to submit a revised version of the manuscript that addresses the points raised during the review process.    

We look forward to receiving your revised manuscript.

Kind regards,

Angela Lupattelli, PhD

Academic Editor

PLOS ONE

Journal Requirements: 

2. You have not indicated whether ethical approval was waived or necessary for your study. We understand that the framework for ethical oversight requirements for studies of this type may differ depending on the setting and we would appreciate some further clarification regarding your research. Could you please provide further details on ethical oversight of your study. Please clarify whether or why your study is exempt from the need for approval

Reviewers' comments:

Reviewer's Responses to Questions

**Comments to the Author**

1. Is the manuscript technically sound, and do the data support the conclusions?

Reviewer #1: Yes

Reviewer #2: Yes

Reviewer #3: Partly

2. Has the statistical analysis been performed appropriately and rigorously? 

Reviewer #1: Yes

Reviewer #2: I Don't Know

Reviewer #3: Yes

3. Have the authors made all data underlying the findings in their manuscript fully available?

Reviewer #1: Yes

Reviewer #2: Yes

Reviewer #3: Yes

4. Is the manuscript presented in an intelligible fashion and written in standard English?

Reviewer #1: Yes

Reviewer #2: Yes

Reviewer #3: Yes

5. Review Comments to the Author

Reviewer #1: The study aims to compare health care utilization among mothers of children with major congenital anomalies (MCA) to that of mothers of children without MCAs through a twenty year period.

This is a major work based on analysis of registry data, combining three sets of data. The study is conducted in a thorough manner.

The authors have written an immaculate script that is easy to follow. The research meets all necessary ethical preconditions. Conclusions are supported by the data and are presented in an appropriate way.

I have two things I want to get elaborated by authors:

1. I wonder about the high number of ‘singles/divorced/widowed/living alone’. It adds up to 44 and 47 % in the groups with and without CMA, respectively. Is it the case that cohabitants are left out? This is an important point as having a partner is of great importance when it comes to how people live through life's various challenges.

2. Major congenital malformations cover a wide variety of prognoses. Some of these are "relatively straightforward", i.e., surgical corrected without sequelae (e. g. Q793, gastroschisis and Q792 omphalocele (unless combined with chromosome abnormalities)). It would be interesting to study this in more detail by removing this type of MCA from the analyzes to see if the results were changed.

Reviewer #2: Accept as written. No edits. I appreciated that the authors controlled for mood conditions when looking at the use of psychiatric services. Thanks to the authors for a good analysis, discussion of what the results mean, why those results are important, and direction for further research.

Reviewer #3: This paper describes the association between MCA in the offspring and health services utilization of the mother in 20 years following childbirth. The reasons for hospitalizations were not presented, which would have been interesting but may be beyond the scope of the paper. The paper is well written and scientifically sound, however, the interpretations of results may need more caution. Several problems arise in the interpretation of results.

Major comments

1) It is not clear whether mothers with children with MCA had increased rates of health care utilization prior to birth (or prior to pregnancy). It is possible that these women had other health problems that led to both higher risk of congenital anomaly in the offspring and increased use of health services. In fact, Figure 3 suggests elevated rates of health care visits were observed in these women 2 years prior to birth. This would affect the interpretation of your results

2) Page 5: How many women were matched multiple times? (if matching was with replacement)

3) Page 6: “…non-placental complications (including intrauterine hypoxia/birth asphyxia,…” Intrauterine hypoxia and fetal-maternal hemorrhage can be also caused by placental complications (e.g., placental insufficiency, abruption). The dichotomy between ‘placental’ and ‘non-placental’ complications is ambiguous.

4) Another limitation to be discussed in the Discussion: Many statistical comparisons can lead to false positive results.

Minor comments:

1. Figure 3. Labels of Y-axes are not entirely clear. Did you mean, for instance, ‘Average number of visits per person per year’, or ‘Average number of hospitalizations per person per year’ ?

2. Some scales on y-axes (0-2) can be mode granular.

3. Figure 1: How many matched women in the comparison group had missing values or were excluded?

4. While Poisson model is appropriate for this analysis, have you considered Cox model that has better utility for longitudinal data? Was the rate of follow up similar in both groups? If your hypothesis was that care for children with MCA can lead to elevated rates of health care utilization for the mother, regression discontinuity model or interrupted time series may be better tools to test this hypothesis.

5. Table 1 bottom: Mortality by strata defined by the number of child hospitalizations in the first year is tricky to interpret. Relatively larger number infants with MCA die soon after birth or before the hospital discharge than infants without MCA. In contrast, no hospitalizations in the first year generally means good health in the average population. I am not sure why these strata (by the number of hospitalizations in the first year ) analyses were performed - I would drop them from the paper.

6. PLOS authors have the option to publish the peer review history of their article (what does this mean?). If published, this will include your full peer review and any attached files.

Reviewer #1: No

Reviewer #2: No

Reviewer #3: **Yes: **Dr. Sarka Lisonkova

---

## [Author Response · Author response to Decision Letter 0]

9 Nov 2021

We thank the Editor and Reviewers for their thoughtful comments and appreciate the opportunity to revise our manuscript. We believe the manuscript has substantially benefited as a result. 

The major changes in this revision are summarized below:

1. We expanded substantially upon the sensitivity analyses to address concerns about (1) the definition of major congenital anomalies used, which covers a wide variety of prognoses, and (2) the fact that increased health care utilization might have been related to the mother’s own health conditions.

2. We revised the manuscript’s text to emphasize that the limitations of the study include potential sources of bias from a high proportion of mothers without cohabitants, and from many statistical comparisons.

3. We evaluated the methods section carefully. To clarify our cohort selection procedure, we updated the study cohort information about the percentage of mothers who were matched multiple times, and we updated Figure 1.

Our point-by-point responses to the comments are provided in the "Response to Reviewers" document.

---

## [Editor Report · Decision Letter 1]

22 Nov 2021

Mothers of children with major congenital anomalies have increased health care utilization over a 20-year post-birth time horizon

PONE-D-21-20906R1

Dear Dr. Shah,

We’re pleased to inform you that your manuscript has been judged scientifically suitable for publication and will be formally accepted for publication once it meets all outstanding technical requirements.

Kind regards,

Angela Lupattelli, PhD

Academic Editor

PLOS ONE

---

## [Editor Report · Acceptance letter]

29 Nov 2021

PONE-D-21-20906R1 

Mothers of children with major congenital anomalies have increased health care utilization over a 20-year post-birth time horizon 

Dear Dr. Shah:

I'm pleased to inform you that your manuscript has been deemed suitable for publication in PLOS ONE. Congratulations! Your manuscript is now with our production department. 

Kind regards, 

on behalf of

Dr. Angela Lupattelli 

Academic Editor

PLOS ONE